# Decision-related feedback in visual cortex lacks spatial selectivity

Katrina R. Quinn[1], Lenka Seillier[1], Daniel A. Butts[2] & Hendrikje Nienborg[3✉]

Feedback in the brain is thought to convey contextual information that underlies our flexibility to perform different tasks. Empirical and computational work on the visual system suggests this is achieved by targeting task-relevant neuronal subpopulations. We combine two tasks, each resulting in selective modulation by feedback, to test whether the feedback reflected the combination of both selectivities. We used visual feature-discrimination specified at one of two possible locations and uncoupled the decision formation from motor plans to report it, while recording in macaque mid-level visual areas. Here we show that although the behavior is spatially selective, using only task-relevant information, modulation by decision-related feedback is spatially unselective. Population responses reveal similar stimulus-choice alignments irrespective of stimulus relevance. The results suggest a common mechanism across tasks, independent of the spatial selectivity these tasks demand. This may reflect biological constraints and facilitate generalization across tasks. Our findings also support a previously hypothesized link between feature-based attention and decision-related activity.

[1] University of Tübingen, Tübingen, Germany. [2] Department of Biology and Program in Neuroscience and Cognitive Science, University of Maryland, College Park, MD, USA. [3] Laboratory of Sensorimotor Research, National Eye Institute, National Institutes of Health, Bethesda, MD, USA. ✉email: hendrikje. nienborg@nih.gov

The brain excels at flexibly performing a multitude of tasks. This ability likely requires the relevant neuronal circuits to have access to task-relevant contextual information. The communication of such context could be supported through feedback signals to upstream populations in accordance with task demands[1,2]. In the visual system, such feedback has been implicated in the modulation of neurons representing task-relevant variables[2–5]. Current thinking suggests that feedback to the visual cortex mediates context-dependent predictions[6], perceptual learning[7], beliefs for hierarchical Bayesian inference[5,8], expectations[9,10], gating[11], or tagging of the relevant sensory information[12,13] to support downstream processing.

These accounts predict the feedback to be selective, targeting some sensory information over other depending on the context of the task and stimulus[14]. Anatomical evidence supports some selectivity of the feedback connections in the visual system[15,16], but the extent to which this enables the selective targeting of specific subsets of neurons is unknown. In light of the enormous number of ethologically possible tasks and contexts, such task-dependent selectivity could become anatomically costly. Limiting the selectivity of the feedback may also be beneficial by facilitating generalization across different tasks.

Here, we set out to test how flexible the selectivity of this feedback is. Specifically, we explored whether previously reported task-specific modulation by feedback[3] is selective for neurons representing a relevant stimulus in the presence of task-irrelevant stimuli. We extended a widely used visual discrimination paradigm[17] to include both task-relevant and task-irrelevant stimuli at different spatial locations; thus, performance in this task required spatial selectivity in addition to selectivity for visual discrimination. During simple visual discrimination tasks using a single stimulus, visual neurons are typically correlated with an animal's choice, unexplained by the stimulus ("choice correlations")[17,18]. Previous work identified a significant decision-related feedback component of these choice correlations[3,19,20]. Conversely, tasks directing attention to one spatial location over others have identified spatially selective modulation of responses in the visual cortex[21].

In the current study, performance in the task required both aspects of selectivity. Thus, we can measure the spatial selectivity of the decision-related modulation by feedback to see how the additional task demands shaped the feedback. If the decision-related feedback modulates the visual neurons selectively according to their task-relevance, it should not affect neurons representing a task-irrelevant stimulus, and these neurons should hence not be correlated with choice (Fig. 1b, c). This also should be the case if choice correlations only reflected feed-forward effects, cf. ref. 22, in which case the predictions for choice correlations would be identical to those for the selective feedback (Fig. 1e, f). Conversely, if the decision-related feedback is unselective to whether the neurons representing the stimulus are relevant for the task, the neurons should show correlations with choice even when representing a task-irrelevant stimulus (Fig. 1h, i). In fact, it has been hypothesized that decision-related feedback in feature discrimination tasks engages the same neural mechanism as feature-based attention[3,4,23]. Studies examining feature-based attention showed that when a subject's attention was directed to a particular stimulus feature, the response of neurons selective for this feature was increased[24–26]. A defining characteristic of such modulation by feature-based attention is that it is observed throughout the visual field[26–29]. As a consequence, if decision-related feedback is linked to the spatially global feedback of feature-based attention, this predicts that neurons representing a task-irrelevant stimulus are correlated with choice (Fig. 1h, i).

Here, we show that although the animals' behavior is highly spatially selective, the decision-related feedback is not. The lack of selectivity cannot be explained by stimulus effects, behavioral covariates, or stimulus- and task-independent neuronal covariability ("noise correlations"). At the level of simultaneously recorded populations, the representation of choice and stimulus is partially misaligned. These stimulus-choice (mis)alignments are similar whether the stimulus is relevant or not. Our results support the previously hypothesized link between feature-based attention and decision-related activity and reveal a feedback mechanism that may support generalization across tasks.

## Results

To test these predictions, we trained two macaque monkeys to perform a disparity discrimination task on a random-dot stereogram (RDS) while ignoring another RDS (Fig. 1j). The relevant stimulus was cued block-wise, and the irrelevant stimulus was presented in the opposite visual hemifield. It was statistically identical but independent of the relevant stimulus to ensure that it provided no information about the correct choice.

**The animals' behavior is spatially selective**. The animals' psychophysical behavior shows that they learned to successfully ignore the task-irrelevant stimulus (Fig. 1k, m).

This was further verified by "psychophysical reverse correlation" analysis, which was computed using trials restricted to the randomly rewarded no-signal trials. This analysis examines any systematic relationship, on average, between the noise in the stimulus and the animals' choices, by computing a "psychophysical kernel" (see "Methods"). Non-zero values of the psychophysical kernel reveal systematic differences of the noise disparities with choice, as can be seen for the relevant stimulus (Fig. 1l, n, top panels), similar to previous findings[19]. In contrast, the amplitude of the psychophysical kernel for the irrelevant stimulus was consistently around zero, confirming that the irrelevant stimulus did not systematically affect the animals' choices. The results were similar when we selected trials for which both the relevant and irrelevant stimulus had no signal (Supplementary Fig. 1). Additionally, we examined the degree to which the irrelevant stimulus or its interaction with the relevant stimulus influenced the animals' behavior using generalized linear model (GLM) analysis (see "Methods"). This GLM yielded weights for each covariate (the relevant stimulus, the irrelevant stimulus and their interaction). The weights for the irrelevant stimulus and for the interaction between the stimuli were close to zero (Supplementary Fig. 2), which also shows that the irrelevant stimulus had minimal influence on the animals' behavior.

**Visual responses are rate modulated by spatial attention**. Furthermore, as is characteristic of the modulation of visual responses by spatial attention[30], we found a substantially smaller response when the stimulus in the receptive field of a unit was irrelevant compared to when it was relevant (Fig. 1o). This modulation of the neuronal response was very consistent across the populations of visual neurons in V2 and V3/V3a (Fig. 1p, V2 mean AI = 0.14, $n = 703$, $p = 10^{-32}$; V3/V3a mean AI = 0.12, $n = 486$, $p = 10^{-30}$, two-sided Wilcoxon signed-rank test for significant deviation from 0).

**Evidence for spatially unselective decision-related feedback**. We next examined how modulation of neural activity by decision-related feedback depended on whether the stimulus was relevant or irrelevant. The behavioral analyses satisfy a key prerequisite for this analysis, since they show that if the irrelevant stimulus had any effect on the animals' choices such effect was very small. We addressed this question using recordings in mid-level areas V2,

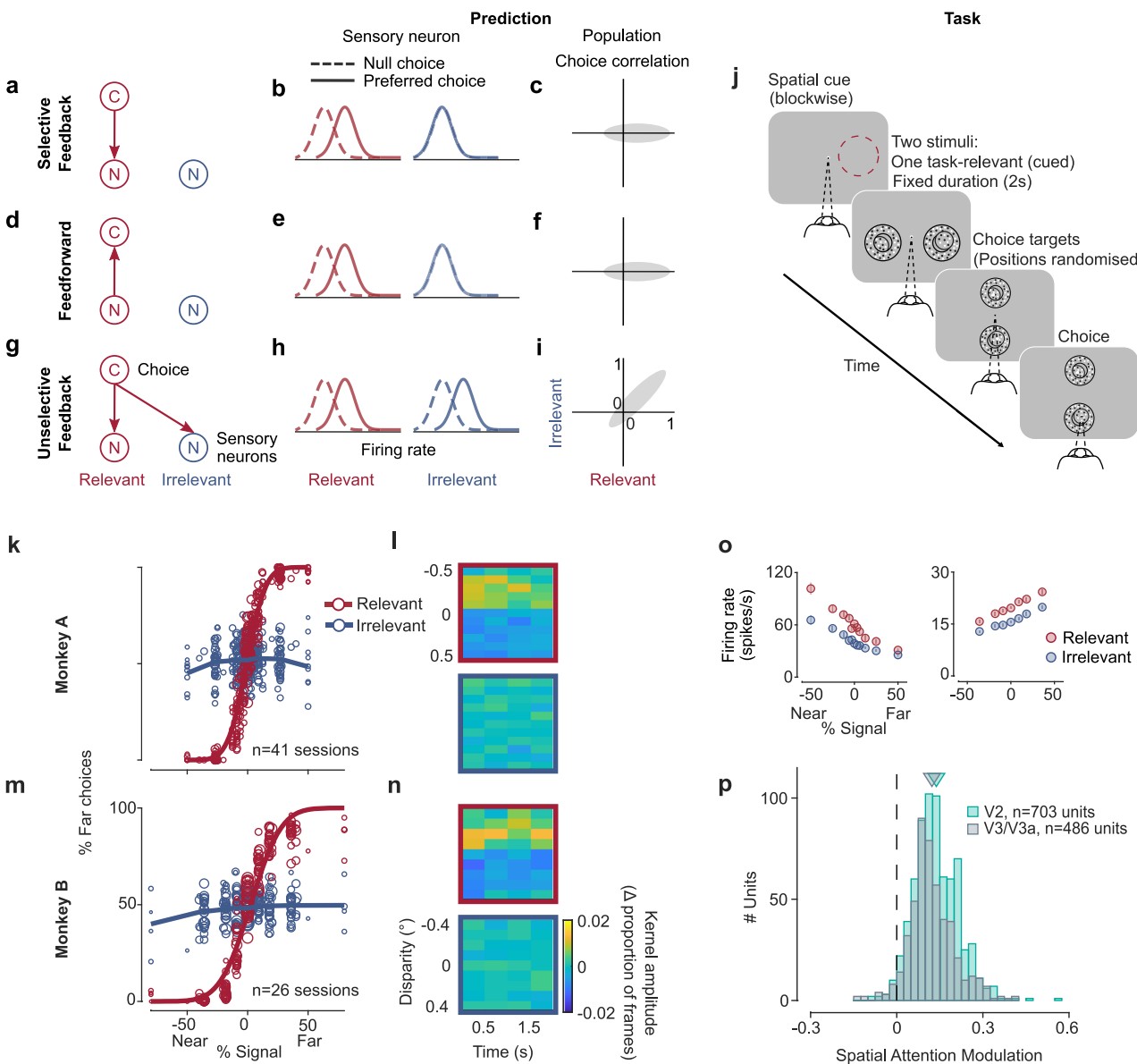

**Fig. 1 Predictions, task, and behavior. a–i** Predictions for choice correlations of visual neurons representing a task-relevant or an ignored, task-irrelevant stimulus. Neurons representing relevant (red) and irrelevant (blue) sensory information (N), or choices (C). Middle column: Firing rate distributions of sensory neurons representing relevant (red) or irrelevant (blue) stimuli, separated for trials on which the observer chose a neuron's preferred (solid) or null stimulus (dashed). Right column: Distribution of choice correlations across a population of sensory neurons representing the relevant (abscissa) or irrelevant (ordinate) stimulus. For selective feedback (**a–c**) or feed-forward processes (**d–f**) choice correlations are only expected for neurons representing the task-relevant stimulus. If feedback is unselective choice correlations are also expected when neurons represent the irrelevant stimulus (**g**, **h**, **i**). For feature-discrimination tasks, spatially global feature-based attention predicts such unselective modulation if these feedback processes are linked. **j** Monkeys performed a disparity discrimination task on a cued stimulus while a task-irrelevant stimulus was simultaneously presented in the opposite hemifield. The relevant hemifield (red circle only for schematic) was indicated at the beginning of each block. **k, m** Psychophysical performance for each session ($n = 41$, $n = 26$ for monkeys A and B, respectively; circle-size proportional to number of trials) for both monkeys and for the relevant (red, cumulative Gaussian fit, psychophysical threshold: 12% (19%) signal for monkey A (B)) and irrelevant (blue) stimulus. **l** Psychophysical kernels (in proportion of frames per 0.5 s time-bin) computed for 0% signal trials (relevant, upper panel; irrelevant, lower panel) as a function of disparity and time for monkey A (trials: $n = 5709$ relevant, $n = 5575$ irrelevant). **n** Same as **l** but for monkey B (trials: $n = 5355$ relevant, $n = 5203$ irrelevant). **o** Tuning curves (spike rate as a function of % disparity signal; error bars show s.e.m., $n = 16–111$ repeats per condition) of two example units (left: V2 unit, monkey A; right: V3/V3a unit, monkey B) when the relevant (red) or irrelevant (blue) stimulus was inside the receptive field. Responses are reduced for the irrelevant stimulus as expected for modulation by spatial attention. **p** Spatial attention index across the population for V2 (cyan, $n = 703$ units, mean = 0.14) and V3/V3a (gray, $n = 486$ units, mean = 0.12).

the earliest site in the visual processing hierarchy for which systematic decision-related activity in disparity-based tasks have been observed[31], and a subsequent processing stage, areas V3/V3a, which has significant disparity selectivity[32]. We computed choice correlations for each unit separately for trials when the

task-relevant (Fig. 2a, x-axis) or task-irrelevant (Fig. 2a, y-axis) stimulus was in its receptive field. Choice correlations quantify the degree to which a unit's firing is correlated with an animal's choice and are closely related (see "Methods") to "choice probabilities" (CPs)[15] (area under the receiver-operating curve

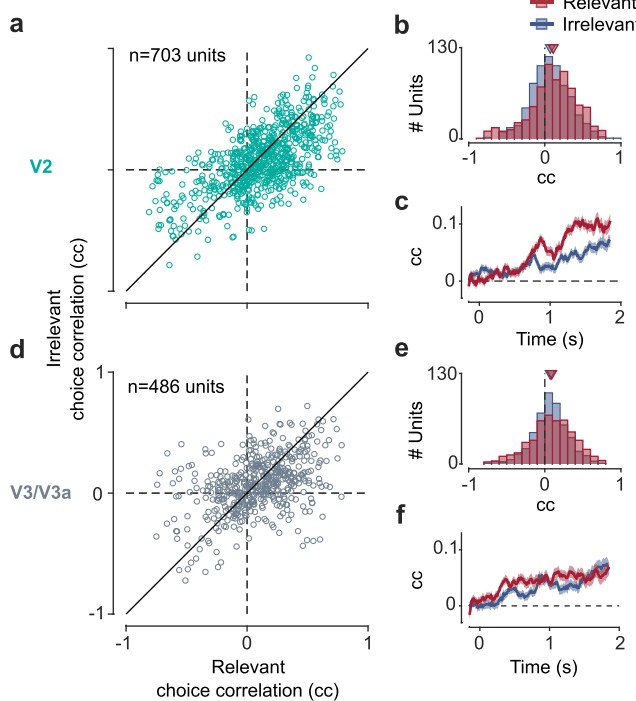

**Fig. 2 Neurons representing the task-irrelevant stimulus are correlated with choice. a** Choice correlations (cc) for $n = 703$ units in V2 when the stimulus in the receptive field was relevant (abscissa) or irrelevant (ordinate) are correlated ($r = 0.61$, $p = 10^{-71}$, two-sided Spearman's rank correlation). **b** Choice correlations for both the relevant (red, mean = 0.11 (area under the receiver-operating characteristic, aROC = 0.55), $p = 10^{-21}$)) and irrelevant (blue, mean = 0.07 (aROC = 0.53), $p = 10^{-12}$, two-sided Wilcoxon signed rank) stimulus were significantly positive and similar although their distributions differed significantly from one another (two-sided Wilcoxon signed rank, $p = 10^{-4}$). **c** Choice correlation (300 ms wide sliding window, 10 ms increment) as a function of time after stimulus onset for the relevant (red) and irrelevant (blue) stimulus. Error bars depict s.e.m. **d–f** Same as **a–c** but for $n = 486$ units in V3/V3a (correlation between relevant and irrelevant choice correlations $r = 0.42$, $p = 10^{-21}$; relevant: mean = 0.09 (aROC = 0.54), $p = 10^{-10}$; irrelevant: mean = 0.07 (aROC = 0.53), $p = 10^{-12}$; difference in distributions, $p = 0.07$).

(aROC)). The choice correlations are signed. That is, positive choice-correlation values mean that a unit has a higher firing rate on trials when the animal chooses this unit's preferred disparity. Conversely, negative choice correlations imply lower firing rates on trials when the animal chooses the unit's preferred disparity. We note that positive choice correlations are expected for e.g. self-reinforcing feedback[5,20], while negative choice correlations would be expected, e.g. for predictive coding[8]. In contrast with what would be predicted in the case of selective feedback (Fig. 1a), we found that units were significantly correlated with choice even when the stimulus in their receptive field was irrelevant to the behavior. On average, the choice correlations for the irrelevant stimulus were positive across the population both in V2 (mean = 0.07, $n = 703$, two-sided sign-rank test for significant deviation from 0: $p = 10^{-12}$; monkey A: mean = 0.09, $n = 543$, $p = 10^{-11}$; monkey B: mean = 0.03, $n = 160$, $p = 0.049$) and V3/V3a (mean = 0.07, $n = 486$, $p = 10^{-12}$; monkey A: mean = 0.10, $n = 315$, $p = 10^{-11}$; monkey B: mean = 0.02, $n = 171$, $p = 0.07$). (Note that decision noise during task performance would lead to worse performance and lower choice correlations, consistent with what we observe in monkey B compared to monkey A.) Across units the choice correlations for the relevant and irrelevant stimulus were

strongly correlated both in V2 (Spearman's rank correlation, $r = 0.61$, $p = 10^{-71}$; monkey A: $r = 0.66$, $p = 10^{-68}$; monkey B: $r = 0.32$, $p = 10^{-4}$) and in V3/V3a ($r = 0.42$, $p = 10^{-21}$; monkey A: $r = 0.38$, $p = 10^{-13}$; monkey B: $r = 0.34$, $p = 10^{-5}$). This finding is incompatible with the predictions for selective feedback (Fig. 1a) or feed-forward accounts (Fig. 1d) but predicted for spatially unselective decision-related feedback (Fig. 1g). It also provides support for the hypothesis that feature-based attention, which is spatially global, and decision-related feedback in our task, engage a linked neural mechanism.

Conversely, the degree to which units were modulated by spatial attention was not related to their choice correlations for either stimulus in V2, and only modestly in V3/V3a (V2: both animals: $r = 0.06$, $p = 0.11$; $r = 0.05$, $p = 0.17$ for Spearman's rank correlation between AI and the cc for the relevant and irrelevant stimulus, respectively; animal A: $r = 0.024$, $p = 0.58$ and $r = 0.018$, $p = 0.68$, respectively; animal B: $r = -0.073$, $p = 0.36$ and $r = 0.012$, $p = 0.88$, respectively; V3/V3a: both animals: $r = 0.16$, $p = 0.001$ and $r = 0.11$, $p = 0.01$, respectively; animal A: $r = 0.16$, $p = 0.01$ and $r = 0.18$, $p = 0.001$, respectively; animal B: $r = 0.12$, $p = 0.13$ and $r = -0.08$, $p = 0.3$, respectively).

The distributions of choice correlations for the relevant and irrelevant stimulus were overall similar, although their pair-wise comparison showed a systematic difference in animal A in V2 (two-sided Wilcoxon signed rank, $p = 10^{-3}$, $p = 0.08$, $p = 10^{-4}$ for animal A, B, and both, respectively), but not in V3/V3a ($p = 0.26$, $p = 0.11$, $p = 0.07$ in animal A, B, and both, respectively). The difference was not explained by the stimulus selectivity of the units (V2: $r = -0.007$, $p = 0.87$; $r = -0.1$, $p = 0.21$; $r = 0.024$, $p = 0.52$. V3/V3a: $r = 0.019$, $p = 0.74$; $r = 0.13$, $p = 0.083$; $r = 0.06$, $p = 0.23$; for animal A, B, and both, respectively, Spearman's rank correlation between the difference of the Fisher-transformed choice correlations for the relevant and irrelevant stimulus and each unit's d-prime). Nonetheless, any such difference could reflect the feed-forward contribution to choice correlations for the relevant stimulus, since the animals used the relevant, but not the irrelevant, stimulus for their decisions. Our animals' behavior relied more strongly on the early part of the relevant stimulus (Fig. 1l, n). A feed-forward component, revealed as a difference between the choice correlations for the relevant and irrelevant stimulus, should therefore be more pronounced early during the trial cf. refs. [19,20]. Yet the difference in choice correlations in our data emerged later (Fig. 2c, f). It therefore suggests that it is not entirely attributable to the feed-forward component and also reflects weakened feedback to the neurons representing the irrelevant stimulus. Since the difference emerged later during the trial although the animals' behavior relied more strongly on the stimulus early during the trial, it occurred at a time when an effect on behavior would likely be weak. Nonetheless, it is conceivable that for tasks that further increase the pressure for spatially selective processing, the decision-related feedback might be more spatially selective.

**Unselective modulation is not explained by the stimulus.** In control experiments and analyses we verified that the choice correlations for the irrelevant stimulus did not result from stimulus-driven, eye-movement, or task-independent effects. First, we found no effect on a unit's firing rate of the stimulus in the opposite hemifield while the animal performed a simple fixation task that would explain the correlations with choice (see example units in Fig. 3b). Indeed, the recorded units showed no systematic difference if the stimulus outside the receptive field had the preferred or null disparity (Fig. 3c). Across the population of the responses for units in V2 and V3/V3a, the proportion of units showing a modulation by the disparity outside the receptive field did not exceed

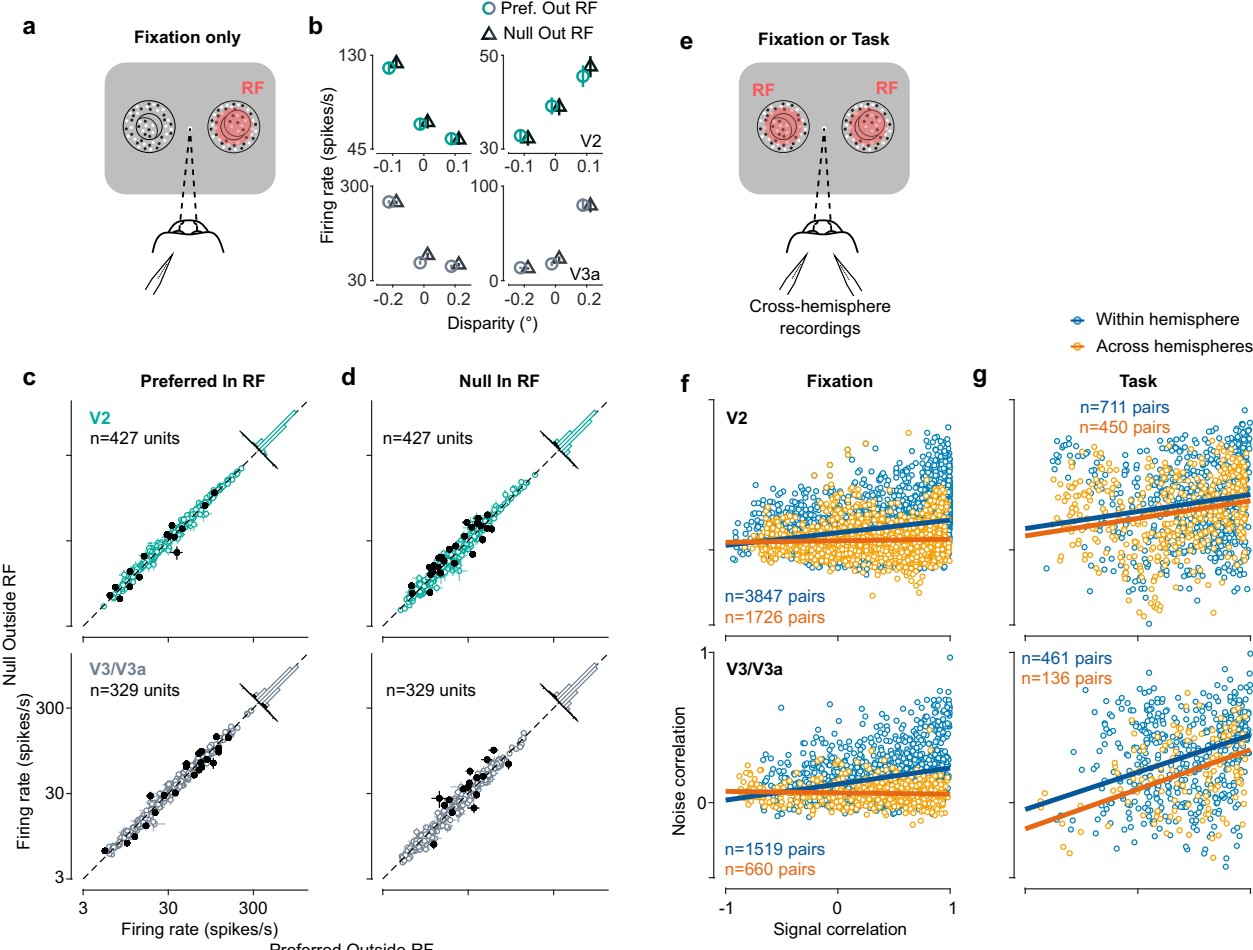

**Fig. 3 Choice correlations for the irrelevant stimulus do not result from the stimulus outside the receptive field, nor from noise correlations in the absence of a task. a** Fixation task with one stimulus presented inside the receptive field, and one stimulus in the opposite hemifield. **b** Disparity tuning curves of four example units (V2, top row; V3/V3a, bottom row) as a function of the disparity of the stimulus inside the receptive field, plotted separately (labels) for the disparity of the stimulus in the opposite hemifield. Data points are horizontally offset for visibility. Error bars show s.e.m. $n = 10$–$40$ repeats per condition. **c** Firing rates to the preferred disparity (top: V2, bottom: V3/V3a) inside the receptive field as a function of the disparity of the stimulus in the opposite hemifield. Histograms show firing rate ratios. Filled data points depict units whose firing rates significantly deviate from unity ($p < 0.05$, two-sample $t$-test). Error bars show s.e.m. **d** as **c** but for the null disparity inside the receptive field. **e** Bi-hemisperic recordings during fixation or task performance. **f** Noise correlations between pairs of units within (blue) and across (orange) hemispheres are plotted as a function of their signal correlation, during fixation and **g** during performance of the task for V2 (top) and V3/V3a (bottom). Note the change in the regression slope within versus across hemispheres during fixation but not during task performance.

chance-level (5% of units for V2, $n = 427$, and V3/V3a, $n = 329$, respectively, two-way ANOVA at 5% significance level).

**Unselective feedback is not explained by behavioral covariates.** Second, our stimulus and task were designed to minimize systematic differences of eye movements with choice by disentangling saccade direction and choice (Fig. 1j, see "Methods"), and to minimize systematic effects of vergence on the neuronal responses. Control analyses (Supplementary Information) show that these do not explain the neuronal correlations with choice. Although we corrected the choice correlations for systematic stimulus differences with choice (see "Methods"), we verified that they were similar when we removed any differences in the randomly generated stimuli between trials (Supplementary Fig. 3).

**Unselective modulation is not explained by task-independent noise correlations.** Third, our findings could not be accounted for by stimulus-independent correlated variability ("noise

correlations"[33,34]) across hemispheres. Such an explanation, e.g. refs. [22,35,36], would stem from previous observations that pairs of neurons with similar stimulus tuning tend to have higher noise correlations than those with dissimilar tuning[34,37,38]. Indeed, we found such a relationship in our data when the animals were fixating and not engaged in the task: within a hemisphere the correlated variability increased with signal correlation (V2: $r = 0.08$, [0.07 0.1], $n = 3847$; V3/V3a: $r = 0.11$, [0.09 0.13], $n = 1519$, type II regression, 95% CI). However, across hemispheres we observed no systematic relationship between correlated variability and signal correlation in V2 ($r = 0.01$, [−0.001 0.02], $n = 1726$) or V3/V3a ($r = −0.01$, [−0.02 0.002], $n = 660$; Fig. 3f, left column). The correlated variability during fixation therefore lacked the structure that would be required to account for choice correlations in a feed-forward way, suggesting that the choice correlations for the task-irrelevant stimulus result from decision-related feedback.

In fact, the presence of choice correlations for the task-irrelevant stimulus implies[39] that neurons in both hemispheres

that share tuning preferences should also receive more similar decision-related feedback compared to inter-hemispheric pairs with different tuning. This would predict that, during task performance, noise correlations are increased between neurons in different hemispheres that have similar tuning (i.e. higher signal correlation). Our results showed exactly this (Fig. 3g; V2: $r = 0.11$, [0.02 0.22], $n = 711$ within hemisphere, $r = 0.12$, [0.07 0.17], $n = 450$ across hemispheres; V3/V3a: $r = 0.25$, [0.14 0.34], $n = 461$ within hemisphere and $r = 0.26$, [0.16 0.37], $n = 136$ across hemispheres). The fact that the structure of the correlated variability changes when the animal engages in the task further supports the finding that this structure depends on task-related feedback[3].

**Analysis of population responses.** To better understand how the structure of the decision-related variability related to the stimulus representation across the population (cf. also ref. [40]), we used a GLM to separately estimate the effect of stimulus tuning and decision-related modulation on neural activity. The model was necessary because the choice and stimulus were often correlated across trials, and the GLM assigns weights according to the best explanation for each effect, which can be statistically distinguished from each other despite their correlation. This model predicted the response of a given neuron (its spike count) on each trial $t$ using the distribution of disparities per trial, the animal's choice, and a drift term (Fig. 4a, see "Methods").

The main parameters of the model used to predict firing rate are the disparity tuning curves for the relevant and irrelevant stimulus, $f_r(x)$ and $f_i(x)$ and the choice weights in each condition $w_r$ and $w_i$. Each tuning curve $f(x)$ is comprised of weights (one weight per disparity $x$), which operate on the histogram of disparities presented over a given trial $n_t(x)$ (Fig. 4a, left), and the choice weights $w$ operate on the choice (Fig. 4a, middle). We also fitted a drift term $d(t)$ that was slowly varying across a session (i.e. one parameter for each cycle of relevant and irrelevant blocks, see "Methods") in order to segregate any non-stationary effects in the recording. For the tuning curves and choice weights, separate weights were fit to trials when the stimulus inside a unit's

receptive field was relevant versus irrelevant, while the same time-varying drift term was applied to all trials.

The best model predictions were accomplished without additional nonlinear mappings (e.g. no need for a spiking nonlinearity link function in the GLM, see "Methods"). Therefore, the model terms that reflect how much each unit was driven by the stimulus and choice are in units of firing rate, and thus can be directly compared across units. This allowed us to examine the degree to which the changes in activity with choice or stimulus were aligned at the level of the population. Specifically, each pattern was represented by a population vector in an $M$-dimensional space, where $M$ is the number of simultaneously recorded units in a given experimental session, and each axis corresponds to the response of one neuron (Fig. 4b). Thus, similar patterns of activity correspond to similar "directions" in population space (relative to the origin) and there would be a correspondingly small angle between them.

**Stimulus-choice (mis)alignments are similar across relevance.** We considered four population response vectors for the changes between the near signal disparity and far signal disparity in the stimulus (see "Methods" for details) when it was relevant (Fig. 4b, **v-stim$_r$**), irrelevant (**v-stim$_i$**), and for the changes in the population response with choice when a relevant (**v-choice$_r$**), or irrelevant (**v-choice$_i$**), stimulus was inside the receptive field of the population. We then computed the angles between these population vectors as a measure of the extent to which the population responses were aligned between the conditions. Across sessions the angles between relevant and irrelevant stimulus vectors ($\Theta_{stim\ r\text{-}i}$) were small, demonstrating that they were well aligned both in V2 (Fig. 4c, top) and V3/V3a (Fig. 4d, top). This is consistent with previously described gain changes associated with spatial attention[41,42] and suggests a roughly uniform effect of spatial attention across the population. But the choice vectors were less well aligned with that of the relevant stimulus ($\Theta_{stim\ r\text{-}choice\ r}$, $\Theta_{stim\ r\text{-}choice\ i}$) in a number of sessions, consistent with recent findings for population recordings in area MT[43].

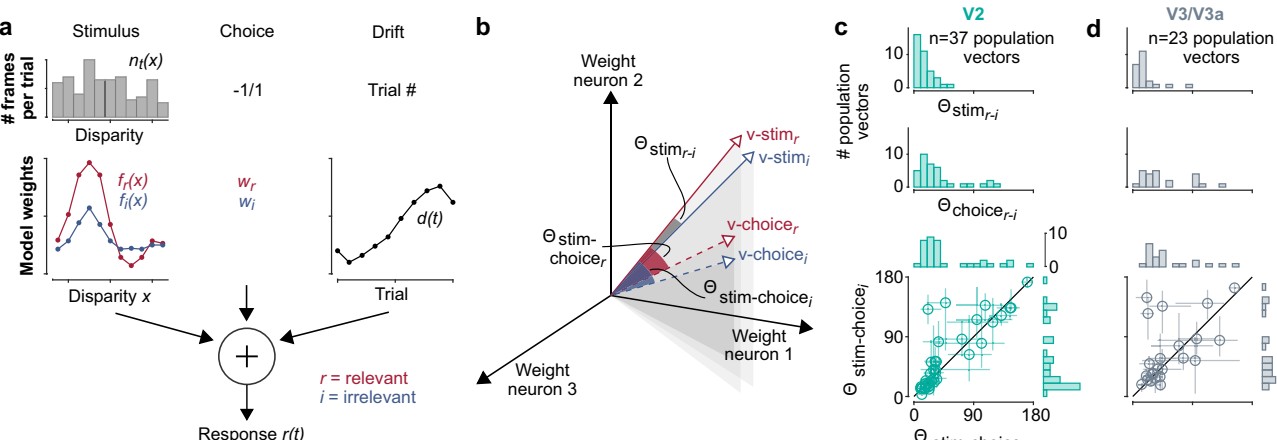

**Fig. 4 Stimulus and choice alignment of the population response in visual cortex. a** The encoding model predicts the spike count in a trial from the stimulus-driven response based on its disparity tuning curve and the disparities presented in that trial (left), the choice during that trial (middle), and a drift term to fit any non-stationarity in the recording over time (right). **b** The stimulus- or choice-driven patterns of neural activity across the recorded neurons can be represented as population vector, with each dimension corresponding to a given neuron's stimulus- or choice-driven weight estimated from the encoding model. The patterns of activity can then be compared as an angle between the population vectors. **c, d** Results for populations in V2 and V3/V3a. Top: The population vectors for the relevant and irrelevant stimulus are well aligned across sessions. Middle: the population vectors for the choices when the stimulus was relevant or irrelevant are broadly aligned. Bottom: The angles between the population vectors for the relevant stimulus (v-choice$_r$ or v-choice$_i$) are correlated ($r = 0.86$, $p = 10^{-10}$, $n = 37$ sessions for V2; $r = 0.51$, $p = 0.01$, $n = 23$ sessions for V3/V3a, two-sided Spearman's rank correlation; error bars are 90% confidence intervals around the median angle, by resampling).

Interestingly, these misalignments between the choice vectors and the relevant stimulus vector were consistent between the relevant and irrelevant choice vectors (Fig. 4c, d, bottom: $r = 0.86$, $p = 10^{-10}$, $n = 37$ for V2; $r = 0.51$, $p = 0.01$, $n = 23$ for V3/V3a). Moreover, the choice vectors ($\Theta_{choice\ r-i}$) were broadly aligned (Fig. 4c, d, middle). This suggests that the decision-related signal affected the population in visual cortex in a similar way, whether this population represented a relevant or irrelevant stimulus, and that this decision-related signal can be misaligned with the stimulus representation. If sensory and non-sensory signals are multiplexed at the level of the sensory population, as previously suggested, e.g. ref. [43], such consistency in the representation of the non-sensory decision signals may facilitate their use by downstream processing.

Additionally, the angles between choice vectors ($\Theta_{choice\ r-i}$) were not correlated with the angles between the relevant versus irrelevant stimulus vectors ($\Theta_{stim\ r-i}$) ($r = 0.15$, $p = 0.37$, $n = 37$ for V2; $r = -0.35$, $p = 0.11$, $n = 23$ for V3/V3a), consistent with the weak (V3/V3a), or absent (V2) correlation between the modulation by spatial attention and choice correlation. This also suggests that the modulation by spatial attention and choice were not prominently coupled in this task.

## Discussion

In summary, we observed substantial choice correlations for neurons representing a task-irrelevant, ignored stimulus, which could not be explained by task-independent covariates or feedforward sensory noise. Rather, these choice correlations require feedback interactions that are roughly similar whether or not the stimulus inside a neuron's receptive field is relevant. From the perspective of the decision-process in this task this is remarkable. Our task was designed to eliminate uncertainty as to which stimulus was task-relevant and analyzing the animals' behavior verified their negligible use of the irrelevant stimulus. Neurons representing this irrelevant sensory information were nearly as correlated with choice as were neurons representing the sensory information that the animals measurably relied on. These findings appear to call into question previously observed systematic links —even if they reflect feedback interactions- between sensory neurons with choice correlations, and the perceptual decision-process, e.g. refs. [17,44–46]. Conversely, the choice correlations for neurons are expected for a mechanism engaging feature-based attention, as previously hypothesized. Our findings here therefore provide support for the hypothesis that the decision-related feedback is linked to the spatially global mechanism engaged in feature-based attention [3,4,23].

However, while a spatially unselective mechanism is beneficial in search or detection tasks that target feature-based attention mechanisms and typically contain spatial uncertainty [24,25], the task used here involved no uncertainty about which location was relevant. Indeed, the measured behavior was highly spatially selective. Any lack of spatial selectivity of the decision-related feedback observed here is therefore not attributable to the demands of the task.

Our findings here extend beyond previous reports (e.g. refs. [47–49]) of unselective task- or decision-related feedback. First, we verified behaviorally that the irrelevant information was ignored by the animals. Second, our task was designed to uncouple the decision-formation from the motor-plan to report the decision.

In this study we explored the selectivity of the feedback once the animals were fully trained on the task. Our results therefore leave open the possibility that the feedback was spatially selective during earlier phases of the training, e.g., to support the animals' learning of the task.

A lack of spatial selectivity of the decision-related feedback could have implications for downstream processing [5,12,20]. It suggests a common mechanism across tasks that is independent of the spatial selectivity that those tasks demand. It also challenges theoretical accounts for the computational role of feedback that require selectivity. The lack of selectivity may result from biological constraints. Assume that the selectivity of the feedback could be increased to target an arbitrary number of stimulus dimensions, and this selectivity is mediated by selective wiring. This would require the number of connections to grow exponentially with each additional stimulus or task dimension. Restricting the selectivity of the modulation by feedback, as observed here, reduces the wiring required for the feedback, and may also facilitate generalization across tasks [50]. Such biological constraints may be key to solving the longstanding puzzle of the computational role of feedback [51] and its implementation.

## Methods

**Animals**. Two adult male rhesus monkeys (*Macaca mulatta*; A, 7 kg; B, 9 kg, both 13 years old; housed in pairs) were implanted with a titanium head-post and two titanium chambers over the operculum of V1 in both hemispheres under general anesthesia. All experimental procedures were approved by the relevant local authority, the Regierungspräsidium Tübingen, Germany.

**Behavioral task**. The monkeys performed a coarse disparity discrimination task on one of two stimuli presented on the screen (Fig. 1j). The task-relevant hemifield was cued at the beginning of each block (50 trials) by three trials during which a single stimulus was presented on the task-relevant side. Once the animal fixated on a fixation point, two dynamic random-dot stereograms (each analogous to ref. [19]) were presented simultaneously (2 s duration), one in each hemifield. Both stimuli were statistically identical but independently varied, and only the task-relevant stimulus was informative about the correct choice. The task was to report whether the central disk of the cued stimulus was protruding ("near") or receding ("far") with respect to its surround. After the stimulus presentation two choice icons (one indicating a "near", one a "far" choice, both at 100% disparity signal and typically horizontally offset towards the cued side by ~1–3°) appeared, whose vertical position (typically 3°–4° above and below the fixation point, held constant within a session) was randomized from trial to trial. This ensured that during the stimulus presentation the motor-command-to-choice mapping was unknown to the animal. The monkeys reported their decision with a saccade to one of the choice icons. Correct choices were rewarded [52] with a liquid reward.

**Electrophysiological recordings**. We recorded extracellular single and multi-unit activity in areas V2 and V3/V3a using multi-channel laminar probes (Plexon, TX, USA; V/S Probes, 24/32 channels, 50–100 μm inter-contact spacing). The position of the probe was changed between sessions, reducing the probability that the same units were sampled across sessions. Eye movements were tracked binocularly using the Eyelink 1000 (SR Research, OTT, Canada) at a sampling frequency of 500 Hz. Neuronal signals were collected using Trellis (Ripple Neuro, UT, USA) interfacing with Matlab (R2014b, Mathworks) via Xippmex (v1.2.1), amplified, digitized, and filtered (250 Hz to 5 kHz) with the Ripple Grapevine System (Ripple Neuro, UT, USA).

*Procedure*. Recording sites were initially mapped using single tungsten in glass electrodes (Alpha Omega, Nazareth, Israel), and selected based on their disparity selectivity and receptive field position.

Probes were inserted into V2 and/or V3/V3a via the operculum of V1, approximately orthonormal to the surface, guided by anatomical MRI scans of the animals' brains, using a microdrive system (NaN Instruments, Israel) with custom-made mounts. V2 was identified as previously described [31] and verified offline based on a shift in a receptive field position and size compared to those in V1. After characterization of the V2 receptive field positions the probes were advanced further and V3/V3a was identified on the basis of shifted and larger receptive fields with respect to those in V2 and clustering for binocular disparity, cf. ref. [32]. Given the previously reported similarity between the disparity selectivity in V3 and V3a [53] we collapsed across recordings in V3 and V3a. Trial-by-trial responses across channels for example sessions in each animal in V2 and V3/V3a are shown in Supplementary Fig. 5.

**Stimuli**. Visual stimuli were back-projected on a screen using a DLP LED Propixx projector (VPixx, Saint-Bruno, Canada; 1920 × 1080 pixels resolution; 30 cd/m² mean luminance; linearized gray values; run at 100 Hz or 60 Hz for each eye) combined with an active circular polarizer (DepthQ, Lightspeed Design Inc., WA, USA; run at 200 or 120 Hz). The monkeys viewed the screen (viewing distance: 103.0 and 97.5 cm in monkeys A and B, respectively) binocularly through passive

circular polarizing filters. Visual stimuli were generated in Matlab (Mathworks) using custom written code based on ref. [54] using the Psychophysics toolbox[55].

Stimuli were circular dynamic random-dot stereograms (RDS, 50% black, 50% white dots, typically 0.08° radius, 50% dot density). They consisted of a disparity varying center (updated at the frame rate of the display, i.e. at 100 or 60 Hz) and a surrounding annulus (1° width, shown at 0° disparity), the size and position of which was determined by the aggregate receptive field of the recorded units (mean RDS center size: 3.6°; mean stimulus eccentricity: 6.3°). On a given stimulus frame, the center dots all had the same disparity, but the disparity could change from frame to frame. Signal disparities (always one near and one far disparity value in each session) were selected to approximately match the disparity selectivity of the majority of the recorded units. Signal frames were interleaved randomly with "noise" frames. The disparity of these noise frames was drawn from a uniform distribution of typically nine values of discrete, equally spaced disparities (symmetrical about 0° disparity, typical values were −0.4°, −0.3°, −0.2°, −0.1°, 0°, 0.1°, 0.2°, 0.3°, 0.4°), encompassing the near and far signal disparity. Signal strength (measured in % signal, signed, where negative values refer to near signal trials, and positive to far signal trials) in a given trial was determined by the proportion of signal to noise frames, and was used to manipulate task-difficulty. For example, −10% signal refers to a trial on which for 10% of the stimulus frames (randomized over time) the central region of the stimulus had the near signal disparity, while the disparity on the remaining 90% of the frames was drawn from the noise distribution. On 0% signal ("no-signal") trials, all frames were drawn from the noise distribution, the correct choice was undefined, and the animal was rewarded randomly on 50% of the trials. The target icons were also RDS but slightly smaller than the stimuli, and always presented at 100% near and far signal.

In a subset of sessions, the random seed used to generate the stimuli was fixed on half of the trials to produce identical stimuli ("frozen noise", see Supplementary Fig. 3).

Disparity tuning curves were measured prior to the behavioral task using identical RDS as used for the task but shown for 450 ms each at 100% signal at changing disparities (typically −1° to 1° in 0.1° increments).

For the control experiments (Fig. 3), identical RDS stimuli as for the task were used with fixed random seeds for a range of signal disparities encompassing the preferred and null disparity of the recorded units.

**Analysis**. Single and multi-unit activity (collectively referred to as units) was sorted offline using the Plexon Offline Sorter (v3.3.5).

*Inclusion criteria*. Only successfully completed trials were included for further analysis. For each unit periods of pronounced non-stationarity were removed. To do so we computed a running 20-trial average spike count, calculated separately for each attention condition. Periods for which this running average decreased below 20% its peak were removed, and the longest continuous segment of included trials was used for further analysis. This resulted in the removal of 8 (0.6%) of the V2 units and 24 (2%) of the V3/V3a units because the remaining data after removing periods of non-stationarity did not meet the minimum number of trials to compute choice correlations (at least five trials with near and far choices, respectively, for the 0% signal stimulus). Only units for which the mean response to the 0% signal stimulus exceeded 4 spikes/s, and for which the d-prime to discriminate the stimuli with the highest signal strength was >0.5 were included. Additionally, the following behavioral criteria had to be met for each unit: the animal's performance had to exceed 70% for the highest % signal stimuli and the bias for 0% signal trials had to be below 75%. For the V2 dataset, out of 1301 units 152 (12%), 266 (20%), and 172 (13%) were excluded because the criteria for minimum firing rate, d-prime or behavior, respectively, were not met. For V3/V3a, out of 1035 units, 123 (12%), 298 (29%), and 104 (10%) were removed for these reasons. Of the included 703 V2 (486 V3/V3a) units 5.5% (3.5%) of trials were excluded due to non-stationarities. Of the included units 18/703 (15/486) were single units in V2 (V3/V3a). The main findings for the single units were qualitatively similar to those for the multi-unit activity (p > 0.35 for all two-sided Wilcoxon rank-sum tests on differences in medians for the choice correlations for the relevant and irrelevant stimulus in either area and across areas), and we therefore collapsed all analyses across single units and multi-units. Non-parametric statistical tests were typically used to avoid relying on assumptions of normality of the data, except for the control analyses in Fig. 3c, d, but the conclusions were unchanged when using non-parametric tests instead.

*Receptive field positions*. For each unit we measured a horizontal and vertical response profile (as described in ref. [56]). These reflect the one-dimensional receptive fields examined along the horizontal axis (using an elongated vertical stimulus, typically a low-spatial frequency sinusoidal luminance grating 0.3–0.5° by 3–5°) or vertical axis (using a horizontally elongated stimulus). These were fit separately with Gaussian functions, and fits were required to explain at least 70% of the variance. The average receptive field position for each area and session (Supplementary Fig. 4) was computed as the average mean of all included fits to the horizontal (x) and vertical (y) response profiles.

*Behavior*. Performance was measured as percent far choices as a function of the relevant or irrelevant stimulus' signed signal strength. Cumulative Gaussians were

fit to the session-averaged performance for the relevant stimulus. Psychophysical thresholds and bias were defined as the standard deviation and mean of these cumulative Gaussians.

*Psychophysical reverse correlation*. Time-resolved psychophysical kernels[57] were computed for 0% signal trials (see also refs. [19,58]) for the relevant and irrelevant stimuli separately. For each of four non-overlapping consecutive time bins (500 ms each), the stimulus was converted to an $n \times m$ matrix ($n$, number of discrete disparity values used for the stimulus; $m$, number of trials). Each entry of this matrix contained the proportion of frames on which a given disparity was presented in this time-bin and trial. The kernel for each time-bin was computed as the difference between the mean matrix across near-choice trials and the mean matrix across far-choice trials. The kernels were averaged across sessions, adjusted for the frame-rate and weighted by the number of trials per session.

*GLM analysis of behavior*. To examine the degree to which the relevant or irrelevant stimulus or their interaction influenced the animals' choices (a far choice is defined as 1) we fit a GLM to the animals' behavior in each session:

$$P(\text{choice} = 1) = \beta_0 + \beta_1 s_{\text{rel}} + \beta_2 s_{\text{irrel}} + \beta_3 s_{\text{rel}} s_{\text{irrel}} \tag{1}$$

where $_\Phi()$ is the cumulative distribution function of the normal distribution, $s_{\text{rel}}$, $s_{\text{irrel}}$, and $s_{\text{rel}} s_{\text{irrel}}$ correspond to the relevant and irrelevant stimulus on each trial (in signed percept signal strength) and their interaction, respectively. We fit weights ($\beta_1$, $\beta_2$, $\beta_3$) to account for the contribution to the animals' choices of each of these covariates as well as the animals' bias ($\beta_0$). To ensure comparability of the resulting weights we normalized (z-scored) the covariates prior to fitting. We used lasso regularization (function *lassoglm* in Matlab; 10-fold cross-validation; see ref. [59]) to avoid overfitting.

*Modulation by spatial attention*. To measure the modulation by spatial attention we compared the mean responses (R, in spikes/sec during the 2 s stimulus presentation) to the 0% signal stimulus when it was relevant versus irrelevant. We quantified the modulation by a contrast (cf.[26]) as a spatial attention index, AI = $(R_{\text{relevant}} - R_{\text{irrelevant}})/(R_{\text{relevant}} + R_{\text{irrelevant}})$.

*Choice correlations*. CPs (signed according to the tuning for disparity of each unit, measured during the fixation tasks outside of the discrimination task) were computed based on the average firing rates for the 0% signal trials. The mean responses, corrected for stimulus-induced effects as described below, for each trial were grouped according to the choice the animal made on a that trial. From the distribution of firing rates when the animal chose a neuron's preferred and null disparity, we computed the aROC, which was defined as CP[17]. CPs can be converted to a Pearson's correlation coefficient "choice correlations", $c_{\text{choice}}$, between the neuronal response and a continuous decision variable as derived previously, i.e. Eq. (S1.6) in ref. [35] and Eq. (8) in ref. [18]:

$$\text{CP} = \frac{1}{2} + \frac{2}{\pi} \tan^{-1} \frac{c_{\text{choice}}}{\sqrt{2 - c_{\text{choice}}^2}} \tag{2}$$

Based on this relationship we converted CPs to choice correlations as done in ref. [60]. Note that Pitkow et al. (2015) used a linear approximation to this quantity: $\text{CP} \approx \frac{1}{2} + \frac{\sqrt{2}}{\pi} c_{\text{choice}} \approx 0.45 c_{\text{choice}} + 0.5$. Because we define the preferred and null disparity based on each unit's disparity tuning, choice correlations are signed with respect to each unit's disparity tuning. That is, positive $c_{\text{choice}}$ values mean that a unit has a higher firing rate on trials when the animal chooses a unit's preferred disparity. Conversely, negative $c_{\text{choice}}$ values imply lower firing rates on trials when the animal chooses the unit's preferred disparity.

To compute choice correlations we first corrected the mean neuronal responses for each trial for fluctuations induced by the stimulus. This is necessary because the psychophysical reverse correlation approach relies on systematic differences on average with choice in the stimulus sequences (Fig. 1l, n). The correction involved two steps. First, we computed a subspace map $s$ (for all 0% signal trials, not separated by choice)[19]: $s$ is a $k$-dimensional vector giving the total number of spikes ($s$) elicited by one frame of a given disparity $x$. A separate subspace map was computed for trials when the stimulus in the receptive field was relevant or irrelevant. Then we summarized the stimulus for each trial by a histogram $n_t^{(r)}(x)$, or $n_t^{(i)}(x)$ (see also below under "Statistical modeling for population analysis") corresponding to the number of frames that the disparity $x$ was presented within the receptive field of the neuron over the trial when the stimulus was relevant or irrelevant, respectively. Second, we calculated the inner product between $s$ and the stimulus histogram $n$ for each trial, which yields the spike count for each trial predicted from the stimulus histogram of that trial. This predicted spike count was subsequently subtracted from the measured spike count on a given trial, which removes the predicted trial-by-trial stimulus-induced fluctuations but not random fluctuations with choice.

The time-courses of the choice correlations were computed analogously but for the firing rate in a sliding (10 ms steps) 300 ms wide window, and corrected for the latency-adjusted stimulus-induced effect.

*Exploring the effect of the stimulus in the opposite hemifield on firing rate.* Units were only included if they exhibited significant disparity tuning ($p < 0.05$ in a one-way ANOVA). We tested for differences in mean firing rate during each 450 ms stimulus presentation for each unit during a fixation task as a function of the stimulus disparity inside the receptive field and in the opposite hemifield using two-way ANOVAs.

*Pair-wise interneuronal covariability.* Pairs of simultaneously recorded units were included if the units were separated by at least 100 μm. Spike-count correlations ("noise correlations") were computed for the average response during the stimulus presentation as the Pearson correlation coefficient between the responses of each pair to an identical stimulus. We required a minimum of 20 presentations per stimulus condition to be included in this computation and then used the average correlation coefficient across stimuli for each pair. Tuning similarity was quantified as "signal correlations"[61] by computing the Pearson correlation coefficient between the tuning curves along the stimulus dimension used for the task (i.e. the mean response to each stimulus as a function of % signed disparity signal) for each pair of units. We quantified the relationship between noise and signal correlation by type II linear regression and significance by resampling (1000 repeats). Our results in Fig. 3f, g were similar when noise correlations were instead computed by first converting the spike counts for each stimulus condition into $z$-scores and then calculated as the Pearson correlation coefficient across $z$-scores.

*Statistical modeling for population analysis.* To distinguish the pattern of activation of the recorded neural populations driven by stimulus and choice—despite their correlation across trials—we used a GLM with an identity link function to predict each neuron's spike count in each trial. The GLM predicted the spike count on a given trial $t$ based on: (1) the number frames $n_t(x)$ that each disparity $x$ was presented in the neuron's receptive field, $n_t^{(r)}(x)$ or $n_t^{(i)}(x)$ depending on whether the stimulus inside the receptive field was relevant or irrelevant, (2) the animal's choice on that trial $c_t^{(r)}$ or $c_t^{(i)}$, again depending on whether the stimulus inside the receptive field was relevant or irrelevant (3) whether the relevant stimulus was in the receptive field or in the opposite hemisphere (see above distinctions between $r$ and $i$); (4) an estimate of the drift $d(t)$ in the firing rates over the recording. An identity link function resulted in the best fits, and as a result each weight has units of firing rate modulation. More complex, nonlinear models[62] had no better model performance than the GLM for predicting the whole-trial spike count:

$$R(t) = \sum_x f_r(x) n_t^{(r)}(x) + \sum_x f_i(x) n_t^{(i)}(x) + w_r c_t^{(r)} + w_i c_t^{(i)} + d(t) \quad (3)$$

where $d(t)$ is the model's drift term estimated using a set of tent-basis functions $\{\xi_j(t)\}$[63] that span all trials (see below) and allow a smooth set of linear model terms, $\{d_j\}$, as follows:

$$d(t) = \sum_j d_j \xi_j(t) \quad (4)$$

Thus, the predictors of the model are defined as follows (for trial $t$):

$n_t^{(r)}(x)$: the histogram of disparities on trials when the stimulus inside the receptive field was relevant, corresponding to the number of frames that the disparity $x$ was presented within the receptive field of the neuron over the trial. Thus, for trials on which the relevant stimulus was presented outside the unit's receptive field, all entries of this vector are set to 0.

$n_t^{(i)}(x)$: the histogram of disparities presented on trials when the stimulus inside the receptive field was irrelevant, corresponding to the number of frames that the disparity $x$ was presented outside the receptive field of the neuron over the trial. Thus, for trials on which the irrelevant stimulus was presented outside the unit's receptive field, all entries of this vector are set to 0.

$c_t^{(r)}$: the choice on trials where the stimulus inside the receptive field was relevant (near = −1, far = +1). If the stimulus inside the receptive field was irrelevant this predictor was set to 0.

$c_t^{(i)}$: the choice on trials where the stimulus inside the receptive field was irrelevant (near = −1, far = +1). If the stimulus inside the receptive field was relevant this predictor was set to 0.

$\{\xi_j(t)\}$: a tent basis (also known as b0-splines) that allows for a smoothly varying drift term (fittable with linear model components) to capture non-stationary aspects of the recording of each neuron not tied to any of the other predictors. There is a basis function for each anchor point $j$, with anchor points chosen with one per cycle of relevant–irrelevant stimulus blocks. The basis function is equal to 1 at the anchor point, and linearly descends to hit zero at the previous ($j$ −1) and next ($j$ + 1) anchor points and is zero everywhere else. Thus, the corresponding model weight $d_j$ gives the value of the offset for the model $d(t)$ at the anchor point, and linearly interpolates at intermediate trials between the values at the anchor points.

Considering the model predictors, the model terms were constrained as follows. For a given trial, linear weights $f(x)$ acting on the histogram of the number of video frames for which the central disc was shown at each disparity $n_t^{(\cdot)}(x)$ yields the average number of spikes evoked by that disparity within the trial. Smoothness of the resulting tuning curve was enforced through regularization using a penalty term on the Laplacian of the weights (e.g., ref. [64]). Choice effects were fit with a single linear weight (one predictor each when the stimulus inside the receptive field was relevant or irrelevant, respectively) acting on a value corresponding to the animal's choice (−1 = near, 1 = far), and thus the model weights $w$ reflected the difference in firing rate resulting from the animal's choice on that trial, again separately for trials when the stimulus was relevant $w_r$ and irrelevant $w_i$. These model terms were fit simultaneously with the "drift term", which captures slow non-stationarities in the firing rate over the recording. As described above, it is fit using parameters that specify the value of the firing rate offset at each anchor point $d_j$, and through the use of tent-basis functions, the value of the offset linearly interpolates between "anchor points" spaced at every period of relevant/irrelevant blocks (roughly every 94 trials; one anchor point per period).

The model parameters were fit simultaneously using gradient descent of the mean-squared error, using custom Python code. By fitting these terms all at once, the GLM could attribute the sources of the modulation to each of these factors, even though some were correlated. Such an encoding approach provides weights for stimulus and choice modulation in units [spikes per trial] that are directly comparable.

*Analysis of stimulus- and choice-driven population activity.* The population vectors were calculated for each recording separately, from the weights of the model fits, with each dimension corresponding to a different neuron. The stimulus vector was based on the difference in weights in response to the near and far signal disparities. That is, for each neuron we computed $f_r(x = \text{near signal}) - f_r(x = \text{far signal})$, which (across neurons) defined a vector in the neuronal population space (Fig. 4b). The population vectors for the irrelevant stimulus were computed analogously from the weights for the irrelevant stimulus. The angles, $\Theta$, between a given pair of population vectors was calculated based on the vector dot-product:

$$\Theta = \cos^{-1} \frac{\boldsymbol{v}_1 \cdot \boldsymbol{v}_2}{|\boldsymbol{v}_1||\boldsymbol{v}_2|} \quad (5)$$

This can be visualized as the angle between lines drawn between the points in population space and the origin (e.g., Fig. 4b).

In addition to the inclusion criteria applied to the main dataset, the population vector analysis excluded units that had a majority of their response (>50%) explained by the drift term in its encoding model. Following this additional screening, only recordings with five or more valid units per area were included in the resulting measurements of population vectors. For the included units the GLM accounted for an average of 52% (V2) and 50% (V3/V3a) of the variance (fivefold cross-validation).

*Citation diversity statement.* Recent work in neuroscience and other fields has identified citation biases suggesting that women and minorities are undercited[65,66]. To increase transparency[67] we here report the citation statistics based on the inferred gender[65] of the first/last authors. Excluding self-citations to the authors of this paper, the references in this paper are 58.6% man/man, 19.0% woman/man, 12.1% man/woman, and 10.3% woman/woman. For comparison, the respective proportions in top neuroscience journals recently reported[66] are 58.6% man/man, 25.3% woman/man, 9.4% man/woman, and 6.7% woman/woman.

**Reporting summary**. Further information on research design is available in the Nature Research Reporting Summary linked to this article.

## Data availability
Source data are provided with this paper as an Excel file (*.xlsx). A portion of the data is deposited here: https://github.com/NienborgLab/Quinn_et_al_2021/Data. The full data reported in this study can be obtained upon reasonable request from the corresponding author.

## Code availability
The custom analysis code is available here: https://github.com/NienborgLab/Quinn_et_al_2021/.

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

## Acknowledgements
We thank Klaus Wimmer, Rich Krauzlis, Ralf Haefner, Bruce Cumming, and Bevil Conway for comments on an earlier draft of this manuscript; Paria Pourriahi and Peter Dicke for technical support; Stephane Clery, Paria Pourriahi, and Katsuhisa Kawaguchi for support with animal training; and Manfred Vernier for animal care. This work was supported by the National Eye Institute Intramural Research Program at the National Institutes of Health (1ZIAEY000570-01), by the European Research Council (ERC: "Neurooptogen") to H.N., the German Research Foundation (DFG), project-numbers 276693517 (TP6), and 211740722 to H.N. and the National Science Foundation (IIS-1350990) to D.A.B.

## Author contributions
Conceptualization: H.N. Data collection: K.R.Q. with support from L.S. and H.N. Analysis: H.N., with support from K.R.Q.; population analysis: D.A.B. Preparation of figures: K.R.Q. Writing: H.N. with support from K.R.Q. and D.A.B.

## Funding

## Competing interests
The authors declare no competing interests.
