## [Peer Review File · Nature Communications]

REVIEWERS' COMMENTS

Reviewer #1 (Remarks to the Author):

The authors have addressed all of my prior concerns with this manuscript. The updated version has clarified the relationship between the results and prior literature and also provides more in-depth analyses of their data, and is much improved.

Reviewer #2 (Remarks to the Author):

The authors have been quite responsive to the previous reviews, and they have addressed my concerns in a serious manner. Acknowledging some of the differences across animals and between main and control conditions does weaken somewhat the impact of the findings, but I do still believe that the core result is pretty solid and interesting. Thus, I have no further concerns for the authors to address.

Reviewer #3 (Remarks to the Author):

I was fairly happy with the paper when it was under review at Nat. Neurosci and I remain fairly happy with it now. The authors have explained the link with feature based attention a bit more clearly and toned down some of the claims about the (lack of) differences between the choice correlations between the relevant and irrelevant stimuli. My main remaining concern is that they haven't really answered my point about whether the non-specific decision-related feedback might have arisen because the monkey was trained at both locations (my general point 1, below).

1) My concern about the limited coverage of related findings was not satisfactorily addressed. I indicated that the Li & Gilbert (2006, 2008, Neuron) demonstrated choice related feedback signals in V1 in a different task, which are attributable to feedback. They demonstrated that the putative feedback signal was absent before learning but appeared when the animals had been trained on a task. The similarities between that result and the present set of findings are substantial:

- In Li & Gilbert the feedback signal was present in blocks of trials in which the stimulus in the receptive field was irrelevant to the task, but only after the task had been trained at both locations.
- Neurons in that task exhibited choice-related activity, because the response magnitude depended on the choice that the monkey made (Li & Gilbert, 2006).
- Li & Gilbert (2006, Neuron) also demonstrated that the feedback effect is most pronounced if the animal uses the stimulus to perform the task. I therefore disagree with the authors' statement in the rebuttal "Indeed, the stimulus and task used by Li et al. (2008), i.e. contour detection in line segments, make it unlikely that it was ignored even when the animal was not asked to perform it. Once segmentation has been successfully learned, it is difficult to unlearn, as evidenced by Gestalt phenomena such as e.g. the Dalmatian dog.)" Vice versa, it is also conceivable that the monkeys of the present study may also have perceived the depth percept on the irrelevant side. Note that I am not denying the novelty of the new results, I simply believe that they could be better positioned in the context of existing literature. It would be helpful if the authors discuss the similarities and differences with Li & Gilbert.
- A related observation has been made by Mirabella et al. (Neuron 2007) showing that choice related feedback goes to V4 neurons in manner that depends on how stimuli of different dimensions (there color and shape) map onto responses, i.e. on the training history. Based on these findings one would predict that the "non-selective choice signal" of the present study would have been weaker or absent had the monkeys been trained at only one location. This is a relevant point to be discussed in the present Ms.

2) Note that my point 2 can be disregarded by the authors (it will not influence my judgement) because it is related to my own work, but I would like to clarify a few misunderstandings.

- In their rebuttal the authors suggest that a global reward-related neuromodulatory signal postulated by the AGREL model and its successors (Rombouts et al. 2015, Plos Comput Biol; Pozzi et al, 2020, NeurIPS) would not account for the neuronal correlations with choice. However, that is a misunderstanding of the model, which uses a global neuromodulator signal for training, but gives rise to highly selective feedback connections which would predict choice-related feedback to the representation of the stimulus, i.e. a boost of activity of those neurons that supported the choice, as is observed here (see Rombouts et al. 2015, Visual Cognition). The degree to which decision related feedback will go to sensory neurons will depend on the training history and will be stronger for locations that have been relevant at some point in time.

Pieter Roelfsema

We thank the reviewer for the remaining comments and the editors for giving us the opportunity to revise our manuscript.

I was fairly happy with the paper when it was under review at Nat. Neurosci and I remain fairly happy with it now.

We appreciate the overall positive evaluation of the paper.

The authors have explained the link with feature based attention a bit more clearly and toned down some of the claims about the (lack of) differences between the choice correlations between the relevant and irrelevant stimuli. My main remaining concern is that they haven't really answered my point about whether the non-specific decision-related feedback might have arisen because the monkey was trained at both locations (my general point 1, below).

1) My concern about the limited coverage of related findings was not satisfactorily addressed. I indicated that the Li & Gilbert (2006, 2008, Neuron) demonstrated choice related feedback signals in V1 in a different task, which are attributable to feedback. They demonstrated that the putative feedback signal was absent before learning but appeared when the animals had been trained on a task. The similarities between that result and the present set of findings are substantial:

- In Li & Gilbert the feedback signal was present in blocks of trials in which the stimulus in the receptive field was irrelevant to the task, but only after the task had been trained at both locations.

Li and Gilbert's studies are groundbreaking work in perceptual learning. It is correct that the modulation by feedback in their studies emerged with training and was also found when the stimulus was irrelevant to the task: as defined by the experimenter. But what was lacking - understandably because this was not central to their main finding- is a demonstration that the stimulus was irrelevant to the animals.

Considering the stimulus used by Li & Gilbert, it is likely that the animals did not ignore the stimulus when it was irrelevant. If the animals did not understand the irrelevance of the stimulus on those trials and hence did not ignore it, the modulation by feedback they observed is no different from the results for the trained, relevant stimulus.

- Neurons in that task exhibited choice-related activity, because the response magnitude depended on the choice that the monkey made (Li & Gilbert, 2006).

- Li & Gilbert (2006, Neuron) also demonstrated that the feedback effect is most pronounced if the animal uses the stimulus to perform the task. I therefore disagree with the authors' statement in the rebuttal "Indeed, the stimulus and task used by Li et al. (2008), i.e. contour detection in line segments, make it unlikely that it was ignored even when the animal was not asked to perform it. Once segmentation has been successfully learned, it is difficult to unlearn, as evidenced by Gestalt phenomena such as e.g. the Dalmatian dog.)" Vice versa, it is also conceivable that the monkeys of the present study may also have perceived the depth percept on the irrelevant side.

We agree that this is conceivable but the implications differ from those discussed above because we measured the animal's behavior: that is, even if the animals perceived the depth in the irrelevant stimulus, we verified that this did not influence their decisions (measured with psychophysical functions, via psychophysical reverse correlation and GLM). Conversely the feedback we observe here is related to the animals' decisions, and this feedback is evident in neurons that measurably did not support these decisions.

Note that I am not denying the novelty of the new results, I simply believe that they could be better positioned in the context of existing literature. It would be helpful if the authors discuss the similarities and differences with Li & Gilbert.

- A related observation has been made by Mirabella et al. (Neuron 2007) showing that choice related feedback goes to V4 neurons in manner that depends on how stimuli of different dimensions (there color and shape) map onto responses, i.e. on the training history. Based on these findings one would predict that the “non-selective choice signal” of the present study would have been weaker or absent had the monkeys been trained at only one location. This is a relevant point to be discussed in the present Ms.

Mirabella et al. (2007) report modulation for features-conjunctions that share the (learned) categorical motor-command with the feature to be reported. Conversely, our study here was designed to uncouple the motor-command from the decision-formation.

We now position our study with respect to the requested literature on lines 391-398:

‘Our findings here extend beyond previous reports e.g. Li et al. Gilbert (2008), Mirabella et al. (2007) and Steinmetz et al. (2018) of unselective task- or decision-related feedback. First, we verified behaviorally that the irrelevant information was ignored by the animals. Second, our task was designed to uncouple the decision-formation from the motor-plan to report the decision.

In this study we explored the selectivity of the feedback once the animals were fully trained on the task. Our results therefore leave open the possibility that the feedback was spatially selective during earlier phases of the training, e.g. to support the animals’ learning of the task.’

2) Note that my point 2 can be disregarded by the authors (it will not influence my judgement) because it is related to my own work, but I would like to clarify a few misunderstandings. - In their rebuttal the authors suggest that a global reward-related neuromodulatory signal postulated by the AGREL model and its successors (Rombouts et al. 2015, Plos Comput Biol; Pozzi et al, 2020, NeurIPS) would not account for the neuronal correlations with choice. However, that is a misunderstanding of the model, which uses a global neuromodulator signal for training, but gives rise to highly selective feedback connections which would predict choice-related feedback to the representation of the stimulus, i.e. a boost of activity of those neurons that supported the choice, as is observed here (see Rombouts et al. 2015, Visual Cognition). The degree to which decision related feedback will go to sensory neurons will depend on the training history and will be stronger for locations that have been relevant at some point in time.

We thank the reviewer for this additional discussion, and also believe that there is a misunderstanding, which we would like to resolve:

This reviewer writes:

“[...] but gives rise to highly selective feedback connections which would predict choice-related feedback to the representation of the stimulus, i.e. a boost of activity of those neurons that supported the choice, as observed here[...].”

In contrast to the feedback described by this reviewer, our main finding here is that we observed feedback even for neurons that did *not support* the choice, as the animals measurably ignored them.

For this reason, the model and the specific extensions discussed do not seem to be applicable here, but we view the AGREL model as a very elegant framework, which we cite in the introduction.

Pieter Roelfsema